# Alpha Carbonic Anhydrase from *Nitratiruptor tergarcus* Engineered for Increased Activity and Thermostability

**DOI:** 10.3390/ijms25115853

**Published:** 2024-05-28

**Authors:** Colleen Varaidzo Manyumwa, Chenxi Zhang, Carsten Jers, Ivan Mijakovic

**Affiliations:** 1The Novo Nordisk Foundation Center for Biosustainability, Technical University of Denmark, 2800 Kongens Lyngby, Denmark; covama@dtu.dk (C.V.M.); chezha@dtu.dk (C.Z.); cjer@dtu.dk (C.J.); 2Department of Life Sciences, Chalmers University of Technology, SE-41296 Gothenburg, Sweden

**Keywords:** alpha carbonic anhydrase, thermostability, rational engineering, *Nitratiruptor tergarcus*, hydrothermal vent, molecular dynamics simulations, CO_2_ hydration, proton transfer, protein mutagenesis

## Abstract

The development of carbon capture and storage technologies has resulted in a rising interest in the use of carbonic anhydrases (CAs) for CO_2_ fixation at elevated temperatures. In this study, we chose to rationally engineer the α-CA (NtCA) from the thermophilic bacterium *Nitratiruptor tergarcus*, which has been previously suggested to be thermostable by in silico studies. Using a combination of analyses with the DEEPDDG software and available structural knowledge, we selected residues in three regions, namely, the catalytic pocket, the dimeric interface and the surface, in order to increase thermostability and CO_2_ hydration activity. A total of 13 specific mutations, affecting seven amino acids, were assessed. Single, double and quadruple mutants were produced in *Escherichia coli* and analyzed. The best-performing mutations that led to improvements in both activity and stability were D168K, a surface mutation, and R210L, a mutation in the dimeric interface. Apart from these, most mutants showed improved thermostability, with mutants R210K and N88K_R210L showing substantial improvements in activity, up to 11-fold. Molecular dynamics simulations, focusing particularly on residue fluctuations, conformational changes and hydrogen bond analysis, elucidated the structural changes imposed by the mutations. Successful engineering of NtCA provided valuable lessons for further engineering of α-CAs.

## 1. Introduction

The utilization of carbonic anhydrases (CAs) as potential CO_2_ sequestration agents to curb the effects of climate change and reduce greenhouse gas concentrations in the atmosphere has been widely acknowledged and extensively documented in the scientific literature. CAs are zinc metallo-enzymes that are responsible for the hydration of CO_2_ in the presence of water via a proton-shuttling mechanism to produce bicarbonate ions [1,2]. The proton that is shuttled out of the active site comes from a Zn-bound water molecule, which then becomes the hydroxide ion responsible for a nucleophilic attack on CO_2_ held in the hydrophobic pocket [3]. From the eight different classes of CAs (α-, β-, γ-, δ-, ζ-, η-, θ- and ι-CAs), the α-CA has been most studied to date, and its structure is well described [4,5,6,7]. It contains a hydrophobic CO_2_ pocket, zinc-coordinating residues and a hydrophilic region, which mainly consists of residues involved in proton shuttling [8,9,10,11]. Bacterial α-CAs are known to occur as dimeric structures forming an interface between the monomers, which contributes towards thermostability. The enzyme is, however, still active in the monomeric form, as active sites are independent of each other. The β-CAs are functional as dimers or tetramers. The γ-CAs are unique in that they are only active as trimers, sharing active sites between the monomers. This class has been shown to be able to utilize either cobalt or iron; the ζ class can utilize cadmium, and the ι class can use manganese as a cofactor [7,12,13,14,15,16]. In most of the classes, the metal cofactor exists in tetrahedral coordination geometry with three protein residues (either Cys, Glu or His), and the fourth coordination ligand can be a water molecule (H_2_O), small organic molecule or another protein residue, usually an Asp. Native coordination geometries include three His (His_3_) residues and a fourth H_2_O for the α-, γ- and δ-CAs; His_2_Glu(H_2_O) for η-CAs; and HisCys2(X) for the β-, ζ- and θ-CAs, where X can either be H_2_O or Asp [5,6,12,16,17,18,19,20,21].

The application of α-CAs in carbon capture and storage (CCS) technologies including direct air capture and point source post-combustion capture allows for the acceleration of CO_2_ dissolution, which is a slow process in the absence of the enzyme. CO_2_ is introduced into the reactors in a mixture of gases. This necessitates initial dissolution within the solvent before it can undergo further conversion. In post-combustion capture, the gas mixture is released at very high temperatures; thus, the process relies heavily on the thermostability properties of the CA [22]. Therefore, we determined that investigating thermostable CAs in thermophilic organisms, a practice previously recorded in the literature, was a justifiable approach [23,24,25,26,27,28,29]. There have been previous attempts to enhance CA thermostability through protein engineering [30,31,32,33]. Though there is not a universal strategy for engineering proteins to attain specific desired traits, rational protein design, which relies on understanding the structure-function relationship of the protein, has been a successful method for improving enzymes. Numerous computational programs have been created with the goal of predicting mutations that can improve the thermostability of proteins [34,35,36,37,38]. Protein engineering, however, frequently entails a trade-off among different properties. For instance, mutations intended to boost catalytic efficiency may simultaneously decrease thermostability [39].

*Nitratiruptor tergarcus* is a bacterium that was isolated from a hydrothermal field in the Mid-Okinawa Trough, Japan [40]. In a previous work, the thermostability properties of its α-CA (NtCA) were investigated in silico [41], and based on the results, which suggested its thermostability, we decided to center this study on NtCA. In the current work, we made use of rational enzyme engineering to mutate NtCA towards higher thermostability and higher activity. Using site-directed mutagenesis, a total of 23 mutants were derived from NtCA, and these included 13 single and 10 combination mutants. Most single mutants displayed an improvement in thermostability, activity or both. The best-performing mutant overall, D168K, showed the highest activity of 654 WAU/mg after incubation at 30 °C and optimal thermostability, with 69% of this activity retained after incubation for one hour at 80 °C. This is in comparison to NtCA’s residual activity of 29 WAU/mg at 30 °C. Activity after incubation at 90 °C was also achieved for one of the combination mutants. We further probed some of the mutations in silico to explain behavior of the mutants in comparison to the wild type, analyzing residue fluctuations. We also monitored changes in interactions formed by mutant residues as well as those within the protein as a whole.

## 2. Results and Discussion

The enzyme NtCA, originating from the hydrothermal vent bacterium *N. tergarcus,* occurs as a dimer. The residues found in the interaction interface, as well as hotspot residues, which are residues whose mutations destabilize the structure, have been identified in a previous study [41]. The aforementioned work included computational analysis of NtCA, which indicated a thermostability potential for this enzyme. Compared to many other CAs, the enzyme stood out by exhibiting reduced residue fluctuations and correlated residue motions at high temperatures in molecular dynamics (MD) simulations [41]. To investigate the enzyme further, we produced it in *Escherichia coli* and evaluated its thermostability experimentally. CO_2_ hydration activity of NtCA was 29 WAU/mg. Compared to the highly efficient SazCA from *Sulfurihydrogenibium azorense* (7905 WAU/mg), this was relatively low. NtCA displayed a potential for thermostability, with the residual activity after incubation for an hour at 70 °C being approximately 50% relative to that after 30 °C incubation. At 80 °C, however, activity was almost undetectable (Figure 1). We therefore set out to improve both the thermostability and activity of this enzyme.

### 2.1. Mutation Selection Process Yields 13 Single Mutants from Seven Selected Residues

The most important part of rational protein design is an adequate understanding of the protein’s structure and how it functions. The structure and function of α-CAs have been studied in great depth previously [8,9,11,41,42], facilitating the engineering process.

#### 2.1.1. DEEPDDG Analysis Yields Two Surface and Two Interface Residues as Mutation Targets

The dimeric structure for the wild type (WT), NtCA, was initially queried in DEEPDDG [38] for mutations that are potentially stabilizing. For every possible single residue mutation, the resulting change in folding free energy (∆∆G value) calculated by the program helped to predict thermostable mutants. A heatmap of the ∆∆G values for all possible single mutations is shown in Appendix A, and mutations with a predicted ∆∆G above 6.275 kJ/mol were selected for further analysis. These are shown in the heatmap image (Figure 2A). An alignment of NtCA with other well-characterized, thermostable and active bacterial CAs was also used to evaluate conservation of the residue positions suggested by DEEPDDG (Figure 2C). Conserved residues are often either structurally or functionally important. From the DEEPDDG results, we selected four residues for mutation, which are indicated by the red boxes in Figure 2A. This was done by deselecting buried residues that were not involved in the dimerization interface or in the active site. The majority of the residues fell into this category, resulting in the selection of A159, D168, D197 and R210, with the first two located on the surface of the protein and the latter two in the dimerization interface. Their position on the structure is displayed in Figure 2B.

It is worth noting that most of the suggested mutations derived from DEEPDDG were hydrophobic residues. Hydrophobic interactions in a protein have been proven to bring about rigidity by the reduction of flexible regions, which is an important criterion for increasing thermostability. However, a hydrophobic surface may increase the risk of protein aggregation, which in turn decreases activity or renders the enzyme inactive. Considering this, for the selected surface residues from DEEPDDG results, the hydrophobic mutations A159F and D168P were added to the mutation pool, and we also introduced the hydrophilic mutations A159D and D168K in an attempt to increase solubility of NtCA. In the alignment, most of the sequences had a Pro in position D168, supporting our selection from DEEPDDG results. In the case of position A159, the alignment suggested that both hydrophobic and hydrophilic residues are tolerated in that position.

Residue R210 is located in an interesting place in the α-CAs. It is found in between two CO_2_ pocket residues, V209 and W211, but is rotated away from the pocket such that it contributes to the interface. DEEPDDG predicted an increased stability by mutation of this residue to the hydrophobic residue Leu. Due to its proximity to the CO_2_ binding pocket, it might lead to a more hydrophobic pocket, which might additionally lead to an improvement of CO_2_ binding. This mutant was therefore both catalytic and stability-oriented. It is also worth noting that SazCA had a Leu in this position as well. R210K was also included in the mutant pool as the hydrophilic counter-mutant of R210L.

#### 2.1.2. Three Catalytic Pocket Residues Are Selected Resulting in Five Different Single Mutants

In addition to mutations selected for improving stability, residues that form the catalytic pocket were considered for mutation in an attempt to improve activity. From the catalytic pocket, the residues N88, K138 and Y144 were considered. N88 is an important residue involved in the formation of a water network in the catalytic pocket, and although it is conserved amongst the bacterial α-CAs, it was mutated to hydrophilic Lys as well as His. K138 and Y144 are located at the top of the pocket outside the active site and are not actively involved in the CO_2_ reaction. As part of the catalytic pocket, however, these residues are important for its structure, and chemical characteristics are thus considerable targets, especially when trying to adapt characteristics of more active and stable CAs. From the alignment, mostly hydrophobic residues are seen in position K138 in other CAs. We therefore chose to mutate the residue to a Leu (K138L), as observed in CmCA and TaCA. For Y144, either an Ile or Leu is observed in other CAs, and both residues were added to the mutation pool.

All 13 single mutations chosen are listed in Table 1, and their positions are illustrated in Figure 2B.

### 2.2. Characterization of Single Mutants Reveal the Majority to Exhibit Improved Catalytic Efficiency

To evaluate the mutations experimentally, we first produced the mutant enzymes in *E. coli*. Of the 13 mutant proteins evaluated, all were active, and only two exhibited an activity similar to or lower than the parent enzyme (29 WAU/mg). The highest increase in activity was seen in surface mutant D168K, with a residual activity of 654 WAU/mg (Figure 3). The hydrophobic mutation for this residue, D168P, effected a much lower improvement (61.8 WAU/mg). The activity for R210L increased as predicted, to 562 WAU/mg, and mutant R210K had more than 11-fold the activity of the wild type (WT) (343 WAU/mg). Activity for D197L decreased slightly compared to the WT at 30 °C to 27 WAU/mg, but interestingly, for D197E (122 WAU/mg), it was more than four times higher.

Thermostability profiles and MD analyses for the four most improved mutants along with their alternative counter-mutants are presented in the following subsections. Static structure analysis in PIC was performed for all hydrophobic and ionic interactions, and MDs were used to probe RMSD, RMSF and hydrogen bond analysis for the selected mutants. All other thermostability profiles are presented in the Appendix A.

### 2.3. Trajectory Analyses of MD Simulations of Mutants Relate Changes in Residue Interactions and Behaviour upon Mutation to Thermostability Profiles

#### 2.3.1. The Most Efficient Mutant, D168K, Maintains High Activity up to 80 °C

The introduction of mutation D168K resulted in the highest improvement in residual activity of NtCA at 30 °C, over 22-fold. This activity was steadily maintained up until 70 °C and with only a slight decrease at 80 °C (Figure 4A). Negligible activity was, however, observed after incubation of the enzyme for 1 h at 90 °C. We hypothesize that this mutation could have brought about stability to the active site, which seemed unaffected by temperatures until 80 °C. In RMSD analysis (Figure 4B), we observed that its structure had the lowest conformation deviation from the initial structure during the simulation, and it quickly attained equilibration. Interestingly, the presence of the D168K mutation resulted in a more stable N-terminus in both chain A and B, showing lower residue fluctuations compared to the WT and D168P (Figure 4C). This was peculiar, because interface hydrogen bonding was generally decreased, including a reduction in hydrogen bonds in the N-terminus of this mutant (Appendix A). New interactions formed by mutant residue K168 include intra-subunit hydrogen bonds with Y166, Q175 and N87.

Although D168P was not nearly as active as D168K, its thermostability was quite impressive. This mutation probably introduced some rigidity in the structure, which was loosened up by the application of heat. In the static structure analysis from PIC, the hydrophobic mutant residue P168 was observed to form new intra-subunit hydrophobic interactions with residues V85, I92, L167 and P169 in its vicinity. These appear to replace intra-subunit hydrogen bonds formed by the original residue, D168, with V85, Q175 and H173, as well as an ionic interaction with H173, all of which are not detected in the mutant.

#### 2.3.2. Mutant R210K Is Activated by Increasing Temperatures up to 70 °C and Exhibits Highest Activity at 80 °C

An impressive improvement in activity was seen for both mutants R210L and R210K, with a 19-fold and 11-fold higher activity compared to the parent enzyme, respectively. As observed for D168P, incubating R210K at temperatures higher than 30 °C increased the mutant’s activity. After heating for an hour at 70 °C, a 3.8-fold increase in residual activity was observed (Figure 5A) compared to that at 30 °C. Incubation at 80 °C reduced residual activity to approximately 160% relative to that at 30 °C, and no activity was detected at 90 °C.

It was interesting to note a general increase in RMSF across R210L’s structure (Figure 5B). This increase in fluctuations resulted in an increase in short-term hydrogen bonding in the interface, which escalated from 78 bonds in the WT to 115 bonds in R210L (Appendix A). The interface also showed increased fluctuations, particularly H243, E143 and hotspot residue E207 (Figure 5C). H243 in the WT forms interface hydrogen bonds with the target residue R210, but upon its mutation to L, the mutant residue L210 now forms hydrophobic interactions with A244′ instead. L210 also formed intra-subunit hydrophobic interactions with V104, F109 and L145.

#### 2.3.3. A159F Displays Exceptional Thermostability up to 70 °C Supported by Stable Structure RMSD and Reduced Residue Fluctuations

Thermostability for mutant A159F was impressively increased, as predicted by DEEPDDG, where the CA retained more than 100% activity after 1 h at 70 °C and 50% at 80 °C relative to activity at 30 °C (Figure 6A). Negligible activity was observed after incubation at 90 °C for A159F. Phe is a much larger and more hydrophobic residue compared to Ala; thus, more hydrophobic interactions were formed with this mutation. Query of the static structures in PIC revealed that in addition to F159–F186 hydrophobic interactions that were found in the WT (A159–F186), the following hydrophobic interactions were also newly formed in the mutant: F159–Y114, F159–V163, F159–F181 and F159–P183. The formation of this hydrophobic cluster is considered in this study to have induced thermostability for this mutant, and this phenomenon has been previously observed [43].

The N-terminus was observed to exhibit a reduction in fluctuations compared to the WT (Figure 6C) as well as the catalytic pocket residues, including Y33, N88, H90, F200, T201 and T202. CO_2_ binding pocket residues F200, T201 and T202 are found on a loop (cyan arrows in Figure 6C) that had a decrease in RMSF upon mutation of A159 to Phe in both chains, possibly resulting in active site stability. Another loop that was found with decreased fluctuations was S234, N235, N236, R237 and P238 (yellow arrows in Figure 6C). This loop is located just behind the catalytic pocket, and its decrease in fluctuation possibly helped to maintain the pocket structure. From this loop, R237 was involved in several new intra-subunit hydrogen bonds that were observed in the static structure of A159F, and these were R237-T201, R237-S59 and R237-Q240. N235 was also observed forming hydrogen bonds with T118, right next to Zn^2+^ coordinating H117. Minimal change was seen in interface RMSF in both mutations. RMSD analysis results were consistent with RMSF analysis, revealing a structure that showed the lowest deviations from the initial structure, exhibiting a constant average range of conformations throughout the trajectory, shown by the very narrow peak closer to the left of the graph (Figure 6B). Hydrogen bonds formed in the interface of the mutants are shown in Appendix A.

Although the A159D mutant had a 6.3-fold higher residual activity compared to the parent enzyme, the thermal stability decreased compared to A159F. The residual activity after incubation at 70 °C was only 29 WAU/mg, and it was completely inactive after incubation at 80 °C. Heating this mutant enzyme to 50 °C increased activity, similar to D168P and R210L/K. Unlike A159F, A159D’s MD trajectory analysis showed it to exhibit two distinct conformations represented by the two peaks in the KDE plot (Figure 6B). It was also seen in the RMSD line graph how the structure deviated further at 50 ns and appeared to equilibrate around the second conformation afterwards. 

### 2.4. Combining Mutations Results in One Mutant Retaining Activity after Incubation at 90 °C

Having identified several single mutations that positively affected activity and stability, we next wanted to assess whether combining some of these mutations would further improve enzyme functionality. We made various combinations of mutations, six mutants with two mutations and four mutants with four mutations (Q1–Q4) (Table 2). All 11 of them contained a mutation of R210 to either Leu or Lys, which individually had a 19- and 11-fold increase in activity, respectively.

Two of the double mutants contained the mutation N88K with the intention of enhancing catalytic activity by adding another proton shuttle in addition to the main proton shuttle, H89.

Most combination mutants were not successful in the improvement of activity and thermostability. The mutant Q4, with A159F, D168P, D197E and R210K, exhibited loss of activity as low as 60 °C (Figure 7). Similarly, the Q1 mutant had very little residual activity after incubation at 60 °C. After incubation at 30 °C, the most improvement in activity was observed for mutant N88K_R210L (Figure 6). It exhibited an improvement in activity compared to the mutant N88K but a lower activity and thermostability than that of the R210L mutant. This double mutant, however, showed the highest residual activity of all mutants in this study after incubation at 90 °C for an hour (22 WAU/mg). Although not much improvement in activity was observed for mutants N88K_R210K, D168K_R210K and Q3, these mutants had the highest activity after incubation at 90 °C at 12%, 25% and 20%, respectively, relative to their activities at 30 °C. Thermostability profiles for all combination mutants are shown in the Appendix A.

## 3. Materials and Methods

### 3.1. Sequence Retrieval, Analyses and Mutation Identification

The protein sequence of *N. tergarcus* CA (NtCA) was retrieved from NCBI [44], and its dimeric 3D structure was modeled using SWISS MODEL [45,46]. The crystal structure from *P. marina* (PDB ID: 6EKI) [29], which has an X-ray diffraction of 2.6 Å, was used as a template with 53% sequence identity to NtCA. It was validated using PROCHECK [47] and Verify3D [48] and was submitted to DEEPDDG [38] to identify residue substitutions that could potentially lead to increased thermostability of the protein. The DEEPDDG server makes use of a neural network-based method, and it mutates each residue individually to each of the 19 amino acid residues aside from the original, then predicts the difference in binding free energy between the mutant and the wild-type protein [38]. The more positive ∆∆G is, the more stabilizing the mutation. A cut-off of ≥6.275 kJ/mol was used as a selection criterium to narrow down the results. The residue mutations suggested by DEEPDDG were screened by taking into consideration the functionality and positions of the residues suggested. All selected mutants were modeled as the wild type as dimers using SWISS-MODEL.

T-COFFEE [49,50] was used to align the NtCA sequence with six other known highly active and thermostable carbonic anhydrases in order to compare residue positions and conservation. Sequences for previously crystallized α-CAs from *S. azorense* (SazCA, PDB ID: 4 × 5 s) [23,51], *S. yellowstonense* sp. YO3AOP1 (SspCA, PDB ID: 4G7A) [52,53], *P. marina* (PmCA1 and PmCA2, PDB IDs 6IM3 and 6EKI, respectively) [28,29,54] and *T. ammonificans* (TaCA, PDB ID:4C3T) [16] were obtained from the Protein Data Bank [55]. The sequence from *C. mediatlanticus* TB-2 [56,57] was obtained from NCBI (Accession number WP 007474387.1).

### 3.2. Bacterial Strains, Plasmids and Mutagenesis

The *E. coli* strains NM522 and BL21(DE3) were used as hosts for cloning and protein synthesis, respectively. Both were grown in low-salt LB medium (10 g/L tryptone, 5 g/L yeast extract and 5 g/L NaCl), shaking at 37 °C. When relevant, kanamycin (50 µg/mL) was added. A synthetic CA gene encoding NtCA without the signal peptide was codon-optimized for *E. coli* and obtained from Integrated DNA Technologies (IDT), Leuven, Belgium, along with relevant primers (Appendix A). It was integrated into the expression vector pETM10 using the Gibson assembly method [58]. Site-directed mutagenesis was achieved by overlap extension using the polymerase chain reaction (PCR). All primers were obtained from IDT.

### 3.3. Protein Expression, Protein Purification and SDS-PAGE

Colonies of *E. coli* BL21(DE3) containing the recombinant genes for NtCA and its mutants were inoculated into low-salt LB medium supplemented with 50 µg/mL kanamycin to a start OD_600_ of 0.02 and cultured at 37 °C with 250 RPM shaking. At an OD_600_ of 0.6, protein synthesis was induced by the addition of 0.1 mM IPTG and 0.1 mM ZnSO_4,_ and the cultures were incubated for an additional 4 h. Cells were harvested by centrifugation (5000× *g* for 15 min), and the pellet was frozen at −20 °C. After thawing, the pellet was resuspended in lysis buffer (25 mM Tris-HCl pH 8.0, 100 mM NaCl, 10% glycerol, 0.2 mg/mL lysozyme) for an hour prior to sonication. Sonication was performed using the Sonics Materials™ Ultrasonic Processor VCX130–220V (Thermo Fisher Scientific Inc., Vantaa, Finland) at 70% amplitude for 3 min (5 s on, 5 s off per cycle) on ice, and the lysate was centrifuged at 4 °C for 10 min at 1000× *g* to remove cell debris. The 6xHis-tagged protein was purified on Ni-NTA resin (Thermo Fisher Scientific Inc.) according to manufacturer’s instructions and subsequently desalted to remove the imidazole using PD-10 columns from Cytiva. All proteins were analyzed by SDS-PAGE to confirm their presence (Appendix A). Protein concentration was estimated using the Bradford protein reagent (Bio-Rad, Hercules, CA, USA) with bovine serum albumin as a standard.

### 3.4. CO_2_ Hydration Assay

CO_2_ hydration activity was measured using the Wilbur-Anderson assay [59], and all assays were performed on ice. A CO_2_-saturated solution was prepared by dissolving dry ice in water for 30 min and used as the substrate. To a Tris-Cl buffer (25 mM, pH 8.4) of 3 mL, 50 µL of enzyme solution was added, and the reaction was started by the addition of 2 mL of the CO_2_ saturated solution. The time for a pH drop from 8.4 to 6.4 was recorded for all reactions including the blank. The activity in Wilbur-Anderson units (WAU) was calculated as follows: (t_0_ − t)/t, where t_0_ is the time it takes for the blank reaction to reach pH 6.4, and t is the time for the enzyme-driven reaction. Activity was divided by protein concentration of the enzyme to obtain WAU/mg.

### 3.5. Thermostability Assays

In order to assess thermostability of NtCA and the effects of mutations on thermostability, enzymes were incubated at 30 °C, 40 °C, 50 °C, 60 °C, 70 °C, 80 °C and 90 °C for an hour and placed on ice before analyzing residual activity using the CO_2_ hydration assay.

### 3.6. Molecular Dynamics Simulations

MD simulations were performed for dimeric assemblies of NtCA parent enzyme and single mutants to investigate the structural causes for differences in activity. Zn^2+^ parameters [60] were inferred using AMBERTOOLS22 [61,62] onto the structures protonated in the H++ server [63]. The AMBERff14SB forcefield [64] was applied, and the structures were minimized using the steepest descent method. NVT (temperature ensemble) and NPT (pressure ensemble) equilibration were executed at 343 K (69.85 °C), and 100 ns simulations were subsequently performed using GROMACS v2021 [65] at the same temperature. Deviation of structures from the initial conformation was measured using root mean square deviation (RMSD) analysis, and residue fluctuations were monitored using root mean square fluctuation (RMSF) analysis with GROMACS. Hydrogen bond analysis was performed using cpptraj [66] in AMBERTOOLS22.

## 4. Conclusions

In this study, we rationally engineered α-CA and NtCA towards both thermostability and catalytic efficiency. After production in *E. coli* and purification, this enzyme was shown to maintain 50% of its activity after an hour of incubation at 70 °C relative to that at 30 °C and was inactive at 80 °C. We identified eight residues for mutation using the vast amount of information on the structure and function of α-CAs, as well as the computational program DEEPDDG. These residues were in regions such as the dimeric interface, the catalytic pocket and the protein surface, and from the eight residues, 13 single mutations were introduced to NtCA. Only two out of the 15 single mutants exhibited similar or reduced residual activity compared to the WT after incubation at 30 °C for an hour. The largest improvement in activity after incubation at 30 °C was seen in the surface mutant, D168K, and the interface mutant, R210L. They showed 69% and 30% relative activity, respectively, after incubation at 80 °C compared to that at 30 °C. Interface mutant R210K displayed both high activity and thermostability compared to the WT. A159F showed impressive thermostability (50% activity at 80 °C relative to 30 °C) with its residues, including the N-terminus, exhibiting lower fluctuations compared to the WT and maintenance of a lower RMSD structure conformation throughout the simulation. Combining the single mutations yielded an interesting mutant, N88K_R210L, which was the only mutant showing a considerable amount of activity at 90 °C. It is also worth mentioning that in the majority of the combination mutants, those containing R210L showed better results compared to their counter-mutants with R210K. Although the mutants in this study still do not match up to the current CA standard in the literature (SazCA from *S. azorense*), they serve mostly as a template for mutation of other CAs rather than being used in CCS technologies.

Overall, rational design of the CA from *N. tergarcus* revealed some interesting prospects for the engineering of other carbonic anhydrases and provided a different angle to mutation selection in this class of enzymes. An important takeaway from this study that may be applicable to other proteins is the integration of various criteria for mutation selection, by which we effectively identified mutations that enhanced protein activity. In future studies, other programs could also be included in filtering various mutations to increase robustness of the selection process. In particular, there might be some benefits to experimenting with mutating residues behind the catalytic pocket that offer structural support, e.g., the loop containing residues 234–238 (NtCA numbering), as compared to the obvious areas such as the active site.

## Figures and Tables

**Figure 1 ijms-25-05853-f001:**
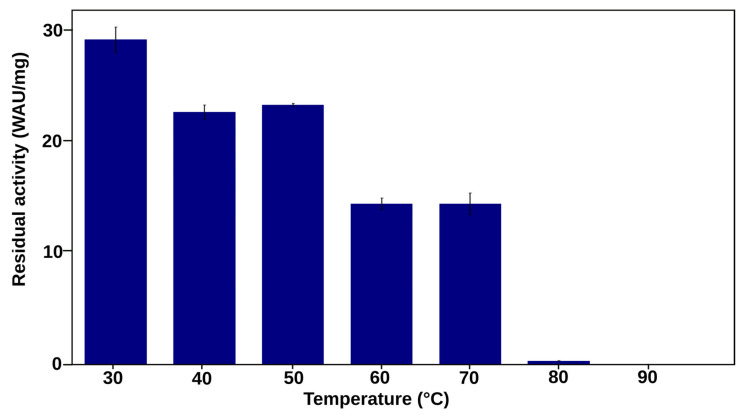
Residual activity of NtCA after 1 h incubation at temperatures in the range 30–90 °C.

**Figure 2 ijms-25-05853-f002:**
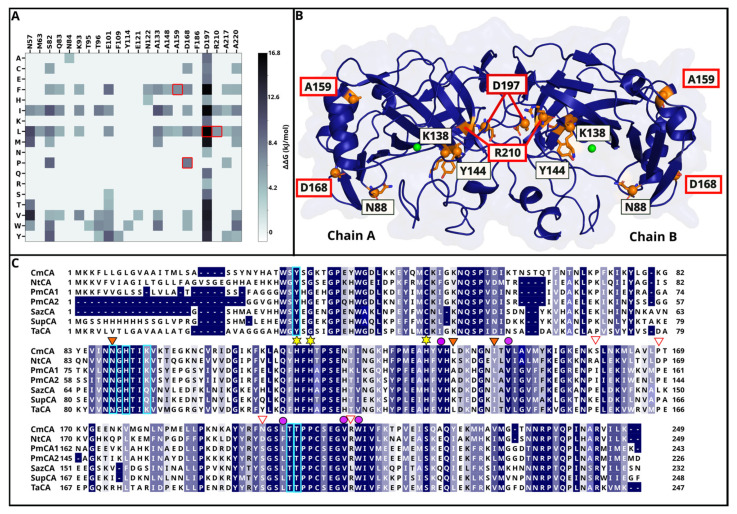
(**A**) Heatmap showing mutations predicted by DEEPDDG to have a ∆∆G > 6.275 kJ/mol. Mutations in the red boxes were chosen for this study. (**B**) Dimeric structure of NtCA showing residues chosen for mutation in both chain A and B. The red boxes show the residues chosen from the DEEPDDG results. (**C**) Alignment of NtCA with CAs from *Caminibacter mediatlanticus* (CmCA), *Persephonella marina* (PmCA1 and PmCA2), *S. azorense* (SazCA), *Sulfurihydrogenibium yellowstonense* (SspCA), *Thermovibrio ammonificans* (TaCA). Triangles show the residues that were selected for mutation, and the residues selected for mutation based on the DEEPDDG analysis are marked with a red triangle. Boxed in cyan are proton transfer residues, yellow-starred residues are Zn^2+^ coordinating histidines, and the purple circles show the CO_2_ binding pocket residues.

**Figure 3 ijms-25-05853-f003:**
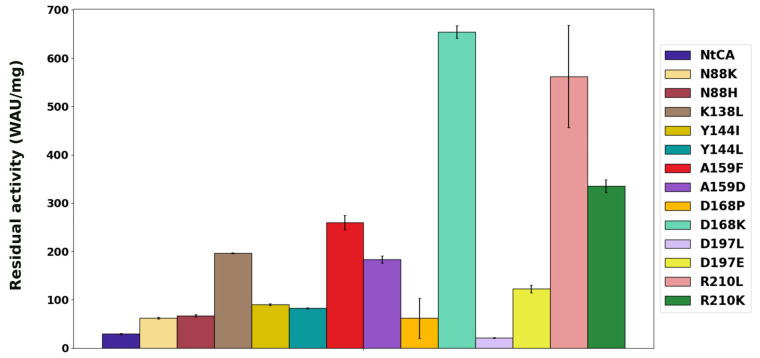
Activities of NtCA and single mutants after incubation for an hour at 30 °C.

**Figure 4 ijms-25-05853-f004:**
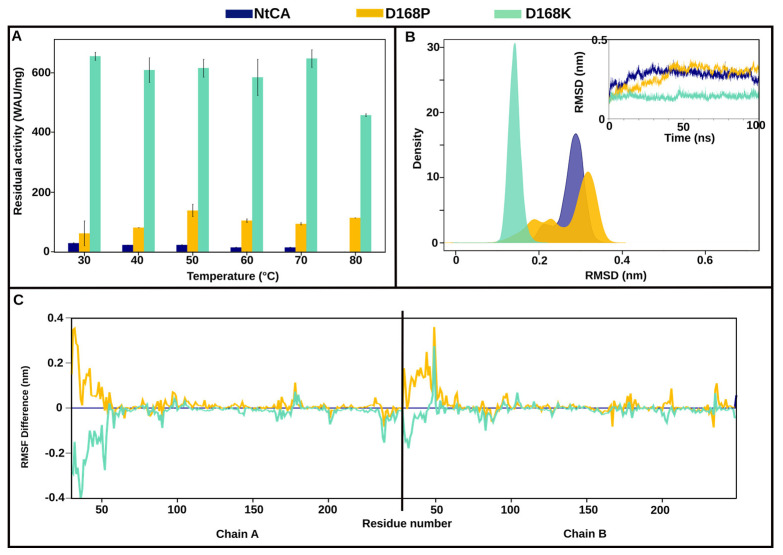
D168P/K. (**A**) Residual activity of NtCA (blue), D168P (orange) and D168K (light green) after incubation for 1 h at 30–80 °C. (**B**) Kernel density estimation graph of root mean square deviation (RMSD) of NtCA, D168P and D168K, with a corresponding RMSD line graph inside calculated over the 100 ns trajectories. (**C**) Root mean square fluctuation (RMSF) line graph of dimeric structures of NtCA, D168P and D168K.

**Figure 5 ijms-25-05853-f005:**
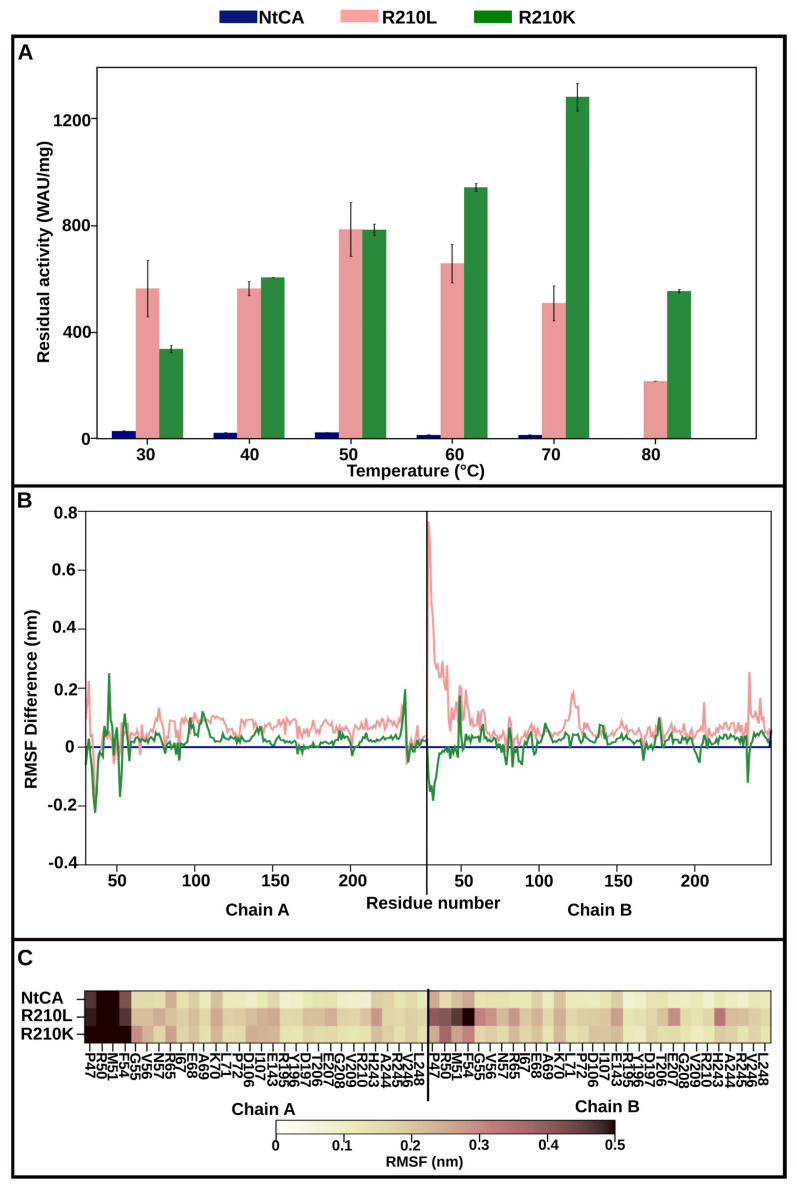
R210L/K. (**A**) Residual activity of NtCA (blue), R210L (pink) and R210K (green) after incubation for 1 h at 30–80 °C. (**B**) The line graph shows the difference of RMSF of residues in R210L (pink) and R210K (green) compared to NtCA (blue). (**C**) RMSF heatmap of the interface residues in the dimeric structures of NtCA, R210L and R210K.

**Figure 6 ijms-25-05853-f006:**
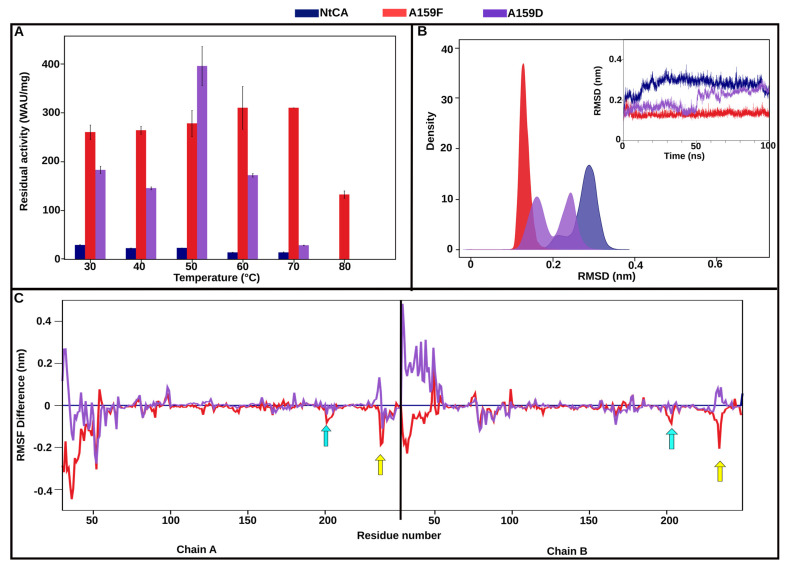
A159F/D. (**A**) Residual activity of NtCA (blue), A159F (red) and A159D (purple) after incubation for 1 h at 30–80 °C. (**B**) Kernel density estimation graph of RMSD, with a corresponding RMSD line graph inside. (**C**) The line graph shows the difference in RMSF of residues in A159F (red) and A159D (purple) compared to NtCA (blue). Yellow and blue arrows are indicative of loops showing a reduction in fluctuations upon mutation to A159F.

**Figure 7 ijms-25-05853-f007:**
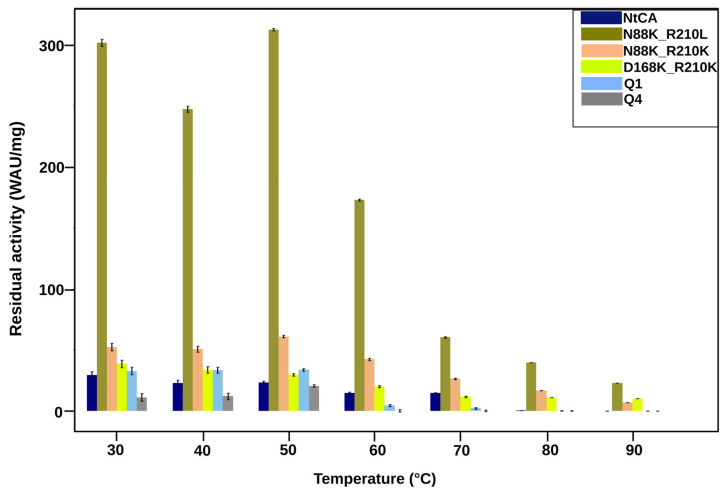
Residual activity of NtCA and some of the combination mutants after incubation for 1 h at 30–90 °C.

**Table 1 ijms-25-05853-t001:** Summary of mutation sites in this study, their position in the 3D structure, the residues they were mutated to and the rationale for mutations.

Mutation Site	Residue Position	Mutant Residue	Rationale
N88	Active site	K H	Catalytic pocket engineering
K138	Close to active site	L	Catalytic pocket engineering
Y144	Close to active site	I L	Catalytic pocket engineering
A159	Surface residue	F D	DEEPDDG/Surface engineering
D168	Surface residue	P K	DEEPDDG/Surface engineering
D197	Interface residue	L E	DEEPDDG/Interface engineering
R210	Interface residue	L K	DEEPDDG/Interface/Active site engineering

**Table 2 ijms-25-05853-t002:** Combination mutations for NtCA for α-CA sequences.

Mutations	Residue Functions	Mutant Name
A159F, R210L	Surface and interface	A159F_R210L
A159F, R210K	Surface and interface	A159F_R210K
N88K, R210L	Active site and interface	N88K_R210L
N88K, R210K	Active site and interface	N88K_R210K
D168K, R210L	Surface and interface	D168K_R210L
D168K, R210K	Surface and interface	D168K_R210K
A159F, D168P, D197L, R210L	2 Surface and 2 interface	Q1
A159F, D168P, D197E, R210L	2 Surface and 2 interface	Q2
A159F, D168P, D197L, R210K	2 Surface and 2 interface	Q3
A159F, D168P, D197E, R210K	2 Surface and 2 interface	Q4

## Data Availability

The data generated in the present study are included in this published article. Protein models are available from the corresponding author upon reasonable request.

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
