# Peer review of "Alpha Carbonic Anhydrase from Nitratiruptor tergarcus Engineered for Increased Activity and Thermostability"

_ijms, 2024, doi:10.3390/ijms25115853_

Round 1

Reviewer 1 Report

Comments and Suggestions for Authors

In this work, the authors present an intriguing computational approach aimed at enhancing the activity and thermostability of Alpha carbonic anhydrase from Nitratiruptor tergarcus. Despite the novelty and the significance of the results, some concepts and results require clarification.

A major point of concern is the lack of clarity regarding the distinction between the effects on activity and stability.  The authors should clarify in the experimental section the different protocols for activity and thermostability assays. If the authors are conducting both activity and thermostability assays using a one-hour time frame, it may lead to ambiguity in the interpretation of results.  Based on the observed text, it appears that the authors are simultaneously evaluating both activity and stability using the activity assay.

To address this issue effectively, the authors should provide clarity in their experimental procedures by explicitly delineating the distinct protocols for activity and thermostability assays. For instance, if one hour is designated for the activity assay, the authors should specify an additional incubation period before conducting the thermostability assay to evaluate the enzyme's resilience to elevated temperatures. By clearly outlining the specific procedures for each assay, including incubation times and conditions, facilitate a better understanding of the observed effects on enzyme activity and stability.

Minor points:

-In the abstract section and at the end of the introduction section, the authors only describe qualitative changes in terms of activity and stability. Perhaps they should attempt to quantify these changes.

-The acronyms corresponding to the names of organisms should be written in italics. (eg. NtCA, SazCA)

-if the term "residual activity" refers to the activity of the enzyme relative to its maximum activity, then "relative activity" would be a more appropriate term

Comments on the Quality of English Language

No comments

Author Response

We thank the reviewer for constructive suggestions. Please find our detailed responses below and the corresponding revisions/corrections highlighted in the re-submitted files. 

Comment 1: In this work, the authors present an intriguing computational approach aimed at enhancing the activity and thermostability of Alpha carbonic anhydrase from Nitratiruptor tergarcus. Despite the novelty and the significance of the results, some concepts and results require clarification. 

A major point of concern is the lack of clarity regarding the distinction between the effects on activity and stability.  The authors should clarify in the experimental section the different protocols for activity and thermostability assays. If the authors are conducting both activity and thermostability assays using a one-hour time frame, it may lead to ambiguity in the interpretation of results.  Based on the observed text, it appears that the authors are simultaneously evaluating both activity and stability using the activity assay. 

To address this issue effectively, the authors should provide clarity in their experimental procedures by explicitly delineating the distinct protocols for activity and thermostability assays. For instance, if one hour is designated for the activity assay, the authors should specify an additional incubation period before conducting the thermostability assay to evaluate the enzyme's resilience to elevated temperatures. By clearly outlining the specific procedures for each assay, including incubation times and conditions, facilitate a better understanding of the observed effects on enzyme activity and stability. 

Response 1:  We have specified in Thermostability procedure (Methods Section 3.5, Line 415-416) that only thermostability was assessed by the CO2 hydration assay after incubation at different temperatures, instead of saying both activity and thermostability. To avoid the confusion in the results, we have adjusted all graphs to only show residual activity (previously specific activity) for each enzyme as it is the activity after incubation at the various temperatures.  

Lines previously describing specific activity have been corrected and are: 104-105; 214-215; 251-254; 273-275; 309-311; 333-334; 337, 340-341; 435-436; 443-448 

Comment 2: In the abstract section and at the end of the introduction section, the authors only describe qualitative changes in terms of activity and stability. Perhaps they should attempt to quantify these changes. 

Response 2: Quantification of activity values has now been mentioned in the introduction (See lines 84-87) 

Comment 3:  The acronyms corresponding to the names of organisms should be written in italics. (eg. NtCA, SazCA) 

Response 3: Since there is no consensus on the acronyms, we prefer to adopt the form previously used for this enzyme (https://doi.org/10.3390/ijms21218066), unless it violates the guidelines of the journal. 

Comment 4: -if the term "residual activity" refers to the activity of the enzyme relative to its maximum activity, then "relative activity" would be a more appropriate term 

Response 4: The ‘residual activity’ which was shown as line plots in the original manuscript has been removed as mentioned in Response 1 to avoid any misunderstanding of the activities. 

Reviewer 2 Report

Comments and Suggestions for Authors

This paper is well written and the topic of mutants is well explained. The experiments carried out are satisfactory, as in the materials and methods section. In my opinion, this is a good paper.

Introduction:

I think the introduction can be improved. You could add something else about the alpha class. For example , the different beetween alpha, beta, gamma and the other class of CA.

Figures:

Figures 4-5-6 must be improved: the superior part of the legend appears cutted. Can you do them again ?

Conclusions:

Can you describe the future prospectives of this new mutants in biotechnology?

Author Response

We thank the reviewer for constructive suggestions. Please find our detailed responses below and the corresponding revisions/corrections highlighted in the re-submitted files. 

Comment 1: I think the introduction can be improved. You could add something else about the alpha class. For example, the different between alpha, beta, gamma and the other class of CA. 

Response 1: Information on other classes has been added. (See lines 46 to 55). All references have been adjusted accordingly. 

Comment 2: Figures 4-5-6 must be improved: the superior part of the legend appears cutted. Can you do them again ? 

Response 2: The figures have been redone to address this problem. (Lines 229, 254, 285) 

Comment 3: Can you describe the future prospectives of this new mutants in biotechnology? 

Response 3: An explanation of how the mutants generated in this study can be exploited in further mutation studies, rather than for direct use in industry has been provided in Conclusion section (Lines 454-456). The general use of CAs in biotechnology (carbon capture technologies) is however outlined in the introduction (Paragraph 2, Lines 57-65). 

Reviewer 3 Report

Comments and Suggestions for Authors

The article by Mijakovic et al is devoted to engineer mutant carbonic anhydrase enzymes with enhance activity and thermostability. The work was done at a high level, well written, the conclusions are substantiated by the results of experiments.

The authors obtained remarkable results with high activities even at 80 degrees Celsius. An excellent result given the fine balance between activity and thermostability. Authors identified key residues of enzyme for mutation using computational tools, and mutations on these regions showed that they have considerable effect in activity or thermostability. The proposed integration of various criteria for the design of mutation selection could lead to more robust selection process in the field.

This is an interesting study, and the manuscript is well written, and presented to high standards. This reviewer believes that the readers of IJMS will enjoy the content of the manuscript and considers it suitable for publication as it is.

Author Response

We thank the reviewer for the positive review.

Reviewer 4 Report

Comments and Suggestions for Authors

This paper described the rational protein engineering of the NtCA through mutating residues in catalytic pocket, on dimeric interface, and on protein surface. Activities of the mutants were measured using Wilbur-Anderson assay on CAs before and after incubation at elevated temperature. The improved in activities and thermal stabilities were explained using molecular dynamics simulation results, notably root mean square deviation and root mean square fluctuation. Overall, it is well written and warrants acceptance. Please address the following:

1.       Please provide SDS-PAGE images in supplementary information, for the reader to make judgement on the purity of the CA enzymes, on which activities were determined.

Author Response

We thank the reviewer for constructive suggestions. Please find our detailed responses below and the corresponding revisions/corrections in the re-submitted files. 

Comment 1 

Please provide SDS-PAGE images in supplementary information, for the reader to make judgement on the purity of the CA enzymes, on which activities were determined.

Response 1: SDS-PAGE gels of partially purified CAs have now been added to Supplementary Information as suggested. 

Reviewer 5 Report

Comments and Suggestions for Authors

I have gone through the manuscript entitled “Alpha carbonic anhydrase from Nitratiruptor tergarcus engineered for increased activity and thermostability”. In this study, authors engineered the α-CA from Nitratiruptor tergarcus (NtCA) using rational design techniques based on structural knowledge. Single, double and quadruple mutants were leading to improvements in both activity and stability. The manuscript is well design and properly written. However, some aspects must be clarified for a better understanding of the topic and, therefore, I suggest accepting this paper in IJMS after a minor revision. 

1. Authors utilized the structure of P. marina (PDB ID: 6EKI) as a template for homology modeling. In materials and methods section more information about the template structure is needed. At least the sequence identity value with NtCA should be provided.

2. As a matter of curiosity, why did the authors choose to minimize the structure using only the steepest descent method? Typically, after employing the steepest descent method, the conjugate gradient method is utilized.

3. For clearer comprehension, Figure 2B should be enlarged or presented as a separate, larger figure. The current size makes it difficult to identify the residues mentioned by the authors.

4. Since residues K138 and Y144 are not actively involved in the CO2 reaction and were not suggested for mutation by DEEPDDG, more information about the rationale behind selecting these positions for mutation should be provided.

5. Legend in figures 4, 5 and 6 appears to be cropped. 

Comments on the Quality of English Language

The overall quality of the English language in the manuscript is acceptable; however, minor editing is recommended to enhance clarity and coherence. Some sentences could benefit from a smoother flow and better organization of ideas.

Author Response

We thank the reviewer for constructive suggestions. Please find our detailed responses below and the corresponding revisions/corrections highlighted in the re-submitted files. 

Comment 1: Authors utilized the structure of P. marina (PDB ID: 6EKI) as a template for homology modeling. In materials and methods section more information about the template structure is needed. At least the sequence identity value with NtCA should be provided. 

Response 1: Sequence identity and X-ray diffraction of template have been added to the methods section (See lines 353-354). 

Comment 2 : As a matter of curiosity, why did the authors choose to minimize the structure using only the steepest descent method? Typically, after employing the steepest descent method, the conjugate gradient method is utilized. 

Response 2 : To our understanding the higher accuracy of the conjugate gradient method is mostly crucial for analyses such as normal mode analysis (NMA). We have routinely used the steepest descent method for MD analyses, and did not change it in this case but we will consider it for future experiments, where we will compare the two methods to see if there’d be a significant difference for our proteins. 

Comment 3: For clearer comprehension, Figure 2B should be enlarged or presented as a separate, larger figure. The current size makes it difficult to identify the residues mentioned by the authors. 

Response 3 : Figure 2B has been enlarged as suggested. 

Comment 4: Since residues K138 and Y144 are not actively involved in the CO2 reaction and were not suggested for mutation by DEEPDDG, more information about the rationale behind selecting these positions for mutation should be provided. 

Response 4 : We have tried to expand upon the rationale behind K138 and Y144 in lines 176-179, mentioning the importance of the formation and characteristics of the pocket, especially when trying to adapt characteristics of more active and stable CAs. 

Comment 5: Legend in figures 4, 5 and 6 appears to be cropped. 

Response 5 : The figures have been redone to address this problem. 

Round 2

Reviewer 1 Report

Comments and Suggestions for Authors

The authors have addressed all my comments